# The extraembryonic serosa is a frontier epithelium providing the insect egg with a full-range innate immune response

Chris G C Jacobs, Herman P Spaink, Maurijn van der Zee*

Institute of Biology, Leiden University, Leiden, Netherlands

**Abstract** *Drosophila* larvae and adults possess a potent innate immune response, but the response of *Drosophila* eggs is poor. In contrast to *Drosophila*, eggs of the beetle *Tribolium* are protected by a serosa, an extraembryonic epithelium that is present in all insects except higher flies. In this study, we test a possible immune function of this frontier epithelium using *Tc-zen1* RNAi-mediated deletion. First, we show that bacteria propagate twice as fast in serosa-less eggs. Then, we compare the complete transcriptomes of wild-type, control RNAi, and *Tc-zen1* RNAi eggs before and after sterile or septic injury. Infection induces genes involved in Toll and IMD-signaling, melanisation, production of reactive oxygen species and antimicrobial peptides in wild-type eggs but not in serosa-less eggs. Finally, we demonstrate constitutive and induced immune gene expression in the serosal epithelium using in situ hybridization. We conclude that the serosa provides insect eggs with a full-range innate immune response.

## Introduction

To combat infection, insects rely on humoral and local immune responses. The humoral immune response is characterized by the massive secretion of antimicrobial peptides into the hemolymph and is mainly exerted by the fat body. Epithelia and hemocytes play the main role in local immune defenses that comprise melanisation, local AMP production, phagocytosis, and encapsulation (*Lemaitre and Hoffmann, 2007*; *Ganesan et al., 2011*; *Davis and Engstrom, 2012*; *Ferrandon, 2013*; *Ligoxygakis, 2013*; *Wang et al., 2014*). The mechanisms regulating these innate immune responses have largely been uncovered with the aid of genetic and molecular studies in the fruit fly *Drosophila melanogaster*. When microbes invade the fly, their released peptidoglycans are sensed by peptidoglycan recognition proteins (PGRPs) and Gram-negative binding proteins (GNBPs) leading to the activation of the main immune signaling pathways. The meso-diaminopimelic acid-type (DAP-type) peptidoglycans of Gram-negative bacteria activate the IMD pathway, whereas the Lys-type peptidoglycans of Gram-positive bacteria activate the Toll pathway. The activation of the Toll pathway is mediated by a proteolytic cascade of serine proteases leading to the cleavage of the cytokine Spätzle, the ligand of the transmembrane receptor Toll. Activation of the immune signaling pathways leads to nuclear localization of the NF-kappaB factors Dorsal, Dif, or Relish that induce antimicrobial peptides (AMPs). Other upregulated genes are prophenoloxidases (proPOs which mediate melanisation) and dual oxidase (DUOX which produces reactive oxygen species).

*Drosophila* has been extremely helpful uncovering those mechanisms, but research in other insects, such as the mealworm beetle *Tenebrio molitor*, has also generated insightful results. The biochemical details of pathway activation, for instance, have mainly been unraveled using this beetle (See *Park et al., 2010* for review). With the availability of tools such as RNAseq and RNAi, more insect species are being established as model organism for innate immunity research (*Altincicek and Vilcinskas, 2007*; *Waterhouse et al., 2007*; *Gerardo et al., 2010*; *Johnston and Rolff, 2013*; *Johnston et al., 2013*; *Zhu et al., 2013*). In particular the red flour beetle (*Tribolium castaneum*) has received much

*For correspondence: m.van.der.zee@biology.leidenuniv.nl

Competing interests: The authors declare that no competing interests exist.

**eLife digest** Insects are among the most numerous and diverse creatures on Earth, and over a million different species of insects have been described. Insects have a hard exoskeleton that protects their segmented bodies, and adult insects and their young are also well protected from pathogens. To fight off infection by bacteria or viruses, these creatures release antimicrobial molecules in the fluid that bathes their internal organs. Insects can also mount a localized immune response that kills off invading microbes.

Most of what scientists have learned about the insect immune system has come from studying fruit flies. While much of the knowledge gained has been applicable to other insects, there is an important exception—fruit fly eggs are incredibly vulnerable to infection. Eggs from other insects are far better protected. In some species, the mother insect protects her eggs either through scrupulous care or by coating them with her own antimicrobial fluids. However, it was unclear if insect eggs could also defend themselves and counter an infection with a strong immune response.

To better understand the immune response in insect eggs, Jacobs et al. studied the eggs of red flour beetles. These beetles are common agricultural pests that eat stored grains and are often studied by scientists in the laboratory. The beetle eggs share a trait with all other insect eggs that is missing from fruit flies and some other flies; the beetle eggs have an extra layer—called the serosa—that envelops the yolk and the developing embryo.

To test whether this extra layer provides immune protection for the egg, Jacobs et al. used a technique called RNA interference to prevent the formation of the serosa. Beetle eggs either with or without a serosa were then pricked with a bacteria-covered object, and Jacobs et al. observed that the bacteria grew twice as fast in the eggs lacking a serosa compared with the eggs that had a serosa.

Next, Jacobs et al. examined gene expression in response to the infection in the eggs. Over 500 genes that are expressed after an infection were identified, and of these genes, 481 were only expressed in eggs with a serosa. Three of these genes, including two that encode antimicrobial molecules, were looked at in more detail, and found to be only expressed within the serosa, indicating that the serosa is the most likely source of the egg's immune response. Importantly, Jacobs et al. found that eggs with a serosa produce the same immune system response as adult insects and concluded that most insect eggs are far from defenseless and are capable of fending off infection.

attention in innate immune studies (*Zou et al., 2007*; *Altincicek et al., 2008*, *2013*; *Roth et al., 2010*; *Contreras et al., 2013*; *Milutinović et al., 2013*; *Zhong et al., 2013*; *Behrens et al., 2014*). Comparative genome analysis has revealed that components of intracellular immune signaling pathways (Toll, IMD, and JAK/STAT) in *Drosophila* are 1:1 conserved in *Tribolium* (*Zou et al., 2007*). The RNAi knockdown technology has shown that the IMD and Toll pathway are largely functionally conserved (*Shrestha and Kim, 2010*; *Yokoi et al., 2012a*, *2012b*). Their activity does, however, not strictly depend on either Gram-negative or Gram-positive bacteria (*Yokoi et al., 2012a*, *2012b*), but this distinction is also not completely black and white in *Drosophila* (*Leulier et al., 2003*; *Leone et al., 2008*). Nevertheless, species-specific family expansion and sequence divergence in the PGRP and AMP families indicate species-specific differences, possibly required for effective recognition and elimination of evolving pathogens (*Christophides et al., 2002*; *Zou et al., 2007*; *Altincicek et al., 2008*; *Park et al., 2010*).

Not only larvae and adults but also insect eggs are constantly threatened by pathogens (See *Blum and Hilker, 2008*; *Kellner, 2008* for review). *Serratia* bacteria, for instance, have been found inside eggs of corn earworms and corn borers (*Bell, 1969*; *Lynch et al., 1976*) and can infect eggs in the laboratory (*Sikorowski et al., 2001*). We have also shown that *Serratia* infection leads to reduced egg survival in the burying beetle *Nicrophorus vespilloides* (*Jacobs et al., 2014*). Maternal investments have been proposed to counter microbial infections. Female medflies, for example, cover their eggs with antimicrobial secretions (*Marchini et al., 1997*) and in the absence of maternal care, eggs of earwigs die of fungal infection (*Boos et al., 2014*). Two studies focusing on transgenerational immune priming, however, have shown that the antimicrobial activity of eggs is of internal origin (*Sadd and Schmid-Hempel, 2007*; *Zanchi et al., 2012*). This is often implicitly interpreted as maternal loading of

antimicrobials into the egg (*Moreau et al., 2012*), but maternal transfer of bacteria to the eggs also leaves zygotic investment as possibility (*Trauer and Hilker, 2013*; *Freitak et al., 2014*). Overall, it is ecologically relevant to gain a better understanding of the immune system in insect eggs.

The zygotic response in *Drosophila* eggs, however, seems poor. It is not until late stage 15, (one of the latest stages in development when ectoderm and trachea have differentiated), that eggs show up to 25-fold upregulation of antimicrobial peptides (*Tan et al., 2014*). This is incomparable to the upregulation in adult flies that is at least an order of magnitude larger. Except for Cecropin (*Tingvall et al., 2001*), stage 11 embryos do not show any induction of antimicrobial peptides and cannot contain an infection of non-pathogenic bacteria, leading to reduced survival (*Tan et al., 2014*). In strong contrast, we have shown that the eggs of *Tribolium* which were not even half way during development could upregulate several AMPs to levels comparable to the adult (*Jacobs and van der Zee, 2013*). This upregulation depends on the serosa, an extraembryonic epithelium that envelopes yolk and embryo (*Jacobs and van der Zee, 2013*). This membrane is present in all insects but was lost in a small group of higher Diptera (the Schizophora) to which *Drosophila* belongs (*Schmidt-Ott, 2000*; *Rafiqi et al., 2008*). Although two maternal extracellular coverings, the chorion and the vitelline membrane, envelop the insect egg, the serosa is the first cellular epithelium surrounding the egg at the interface between the microbe rich external milieu on the one side and the yolk and embryo at the other side. Thus, the serosa could function as an immune competent barrier epithelium. This has been suggested before, as the NF-kappaB factor Dorsal is highly expressed in the presumptive serosa (*Chen et al., 2000*). The absence of the serosa might account for the poor immune response in *Drosophila* eggs.

To gain deeper insights into the role of the serosa, we chose *Tribolium castaneum*, a beetle that possesses a serosa like all non-Schizophoran insects. In this beetle, we can prevent the development of the serosa by parental RNA interference with *Tc-zerknüllt1* (*Tc-zen1*). This technique generates *Tribolium* eggs with an amnion at the dorsal side, but without a serosa (*van der Zee et al., 2005*). At the relative humidity of the air of the laboratory, normal larvae hatch from these eggs (*Jacobs et al., 2013*). As *Tc-zen1* is only expressed in the early serosa (*van der Zee et al., 2005*) and is not expressed anymore by the time the experiments are performed (See discussion), we expect only to find effects that are a consequence of the absence of the serosa. We investigated the growth of bacteria in serosa-less and wild-type eggs, sequenced the whole transcriptome of naive and immune-challenged eggs with and without a serosal epithelium and confirmed constitutive and induced gene expression in the serosa by in situ hybridization. We conclude that the serosa is a frontier epithelium that provides immune competence to the insect egg.

## Results

### Bacteria propagate twice as fast in serosa-less eggs

To examine the influence of the serosa on bacterial growth in infected eggs, and to standardize our infection method, we counted colony forming units (cfu's) directly after infection (t = 0) and 6 hr later (t = 6) (*Figure 1*). We pricked 24–40hr old eggs (i.e. up to half-way during development) with a tungsten needle dipped in a concentrated mix of *Escherichia coli* and *Micrococcus luteus* cultures (see 'Materials and methods'). To determine cfu's, we shortly treated eggs with 0.5% hypochlorite to sterilize the outside. Untreated eggs did hardly contain bacteria that grow on LB agar plates (on an average three cfu's were found). Sterile injury did not increase this number (*Figure 1*, lower lines). In contrast, septic injury introduced on average 53 bacteria into wild-type eggs and 49 into serosa-less eggs. These numbers increased on average to 747 cfu's in wild-type eggs and to 7260 cfu's in serosa-less eggs. When we use the formula $N_{(t)} = N_{(0)}*e^{kt}$, the specific bacterial growth rate $k$ in wild-type eggs is 0.44 hr$^{-1}$, whereas $k = 0.83$ hr$^{-1}$ in serosa-less eggs. This means that bacteria grow twice as fast in serosa-less eggs and suggests that the serosa exerts an immune function.

### RNAseq reveals a full-range immune response in *Tribolium* eggs

To characterize this immune function, we sequenced the whole transcriptome of wild-type eggs, *Tc-zen1* RNAi (serosa-less) eggs, and control RNAi eggs without injury, after sterile injury, and after septic injury (*Figure 2*). The control RNAi consists of an injection of a 500 bp dsRNA derived from a vector sequence without target in the *Tribolium castaneum* genome. For these nine different treatments, three biological replicates were carried out (independent RNAi, independent injury) giving a total of 27 samples (*Figure 2*). Illumina next generation sequencing resulted in over 970 million cDNA

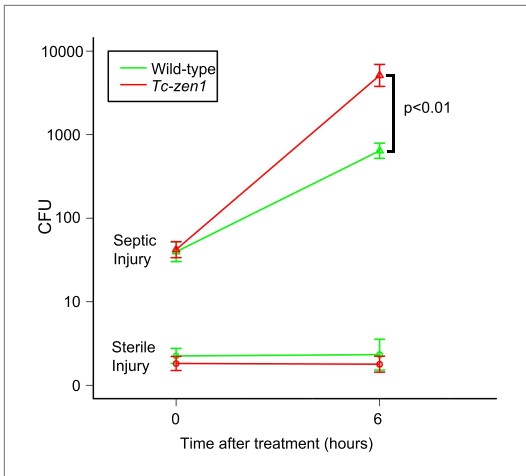

**Figure 1**. Counts of colony forming units (cfu's) after sterile and septic injury. Green lines represent bacterial growth in wild-type eggs. Red lines represent bacterial growth in *Tc-zen1* RNAi (serosa-less) eggs. Sterile injury did not introduce bacteria (lower lines: average of 2 cfu's found at t = 0 and an average of 5 cfu's found at t = 6). Septic injury introduced on average 53 bacteria into wild-type eggs and 49 into serosa-less eggs. These numbers increased to 747 ± 106 cfu's in wild-type eggs (green upper line) and to 7260 ± 1698 cfu's in serosa-less eggs (red upper line) at t = 6. This means that bacteria propagate twice as fast in serosa-less eggs (p < 0.01, as determined by a Pearson's chi-square test). Suspensions of 10 eggs were used per LB agar plate (see 'Materials and methods'), and 10 plates were analyzed per treatment and time point, giving rise to the error bars presented in the graph (standard error).

reads with over 49 billion bp sequence information. Approximately, 72% of the reads could be mapped to *Tribolium* gene models built on the 3.0 genome assembly (*Richards et al., 2008*) (*Supplementary file 1*). We found expression of 14,903 of the total of 16,541 predicted genes, of which 13,464 genes were expressed in wild-type, control, and *Tc-zen1* RNAi eggs and 1440 genes were expressed in a subset of these treatments. These numbers confirm the quality of the deep sequencing data.

First, we identified the immune-responsive genes by determining differential expression of genes between naive eggs on the one hand and sterilely injured eggs or septically injured eggs on the other hand. We only considered genes with at least a twofold change in expression and an adjusted p-value smaller than 0.01. This gave a total of 415 differentially expressed genes in the sterilely injured eggs compared to the naive eggs, and a total of 538 differentially regulated genes in septically injured eggs compared to naive eggs. This shows that *Tribolium* eggs possess an extensive transcriptional response upon infection.

To obtain a global impression of the kind of genes differentially regulated upon infection in wild-type and control RNAi eggs, we assigned gene ontology terms (GO-terms) to all *Tribolium* genes. As no GO-term annotation is available for *Tribolium*, we blasted *Tribolium* genes against *Drosophila* and used the *Drosophila* GO-terms of the best hit. Using the Wallenius approximation (*Young et al., 2010*), we found several highly over-represented GO-term categories with a p-value below 0.001 in both wild-type eggs (*Figure 3A*) and control RNAi eggs (*Figure 3B*). The over-represented categories are mostly immune related. This indicates that our approach does not depend on artefacts generated by pricking eggs (e.g. delayed development) but mainly identifies genes involved in the innate immune response.

To obtain a more detailed analysis of the immune response in wild-type and control eggs, we focused on 368 genes that have been annotated as immune genes (*Zou et al., 2007*; *Altincicek et al., 2013*) (*Supplementary files 4–9*). Of these genes, 78 were differentially regulated in wild-type eggs upon septic injury (*Table 1* and *Supplementary file 2 and 5*), while 95 immune genes were differentially regulated in control RNAi eggs (*Table 1* and *Supplementary file 2 and 7*). This indicates that RNAi itself leads to an increased number of differentially regulated genes upon bacterial challenge but, more importantly, shows that *Tribolium* eggs possess an elaborate immune response. In the following sections, we take a closer look at the exact genes involved in this extensive immune response.

## Recognition of microbes and extracellular signal transduction

Of the 7 predicted peptidoglycan recognition proteins (PGRPs) in *Tribolium* we found significant induction of *PGRP-LA*, *LC*, *SA*, and *SB* (*Supplementary file 2*). Of these PGRPs, *PGRP-SA*, and *SB* were induced over 200-fold (*Figure 4A*, *Supplementary file 2*). Thus, it could be that these PGRPs rather function as effectors digesting Gram-positive bacteria, as shown for human PGRP-S (*Dziarski et al., 2003*). At least *PGRP-SB* shows all the amino acid residues characteristic for catalytic PGRPs (*Kim et al., 2003*). No induction was found for *PGRP-LE* and *LD*. These findings strongly resemble the response of *Tribolium* adults, in which the same PGRPs responded to infection (*Altincicek et al., 2013*). Of the Gram-negative binding proteins (GNBP), we found induction of *GNBP2* and

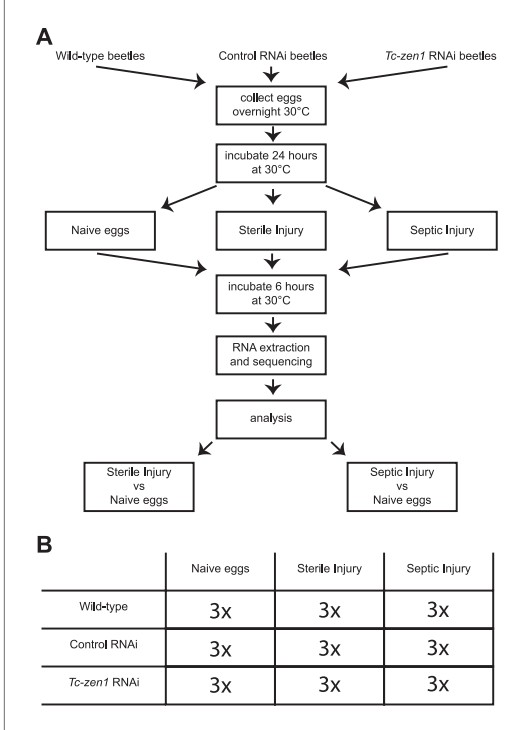

**Figure 2**. Experimental setup. (**A**) We collected eggs from wild-type, control RNAi, and *Tc-zen1* RNAi beetles overnight. These eggs were incubated for 24 hr at 30°C to ensure development of the serosa. Eggs are then maximally 40 hr old, while total developmental time is close to 85 hr at 30°C. Eggs were pricked with a sterile needle (sterile injury), pricked with a mix of *E. coli* and *M. luteus* (septic injury), or remained untreated (naive). They were incubated for another 6 hr at 30°C before total RNA was extracted for RNAseq. To analyze the immune response, the transcriptomes of sterilely injured eggs and of septically injured eggs were compared to naive eggs. This was done for wild-type, control, and *Tc-zen1* RNAi eggs. (**B**) We collected three biological samples for each combination of egg-type (wild-type, control RNAi, or *Tc-zen1* RNAi) and treatment (naive, sterile injury, or septic injury) giving a total of 27 biological samples.

GNBP3 (*Supplementary file 2*). In *Tribolium* adults and in *Drosophila*, however, only *GNBP3* is immune-inducible (*Lemaitre and Hoffmann, 2007*; *Altincicek et al., 2013*), whereas *GNBP1* and *GNBP3* are immune-inducible in *Tenebrio* (*Johnston et al., 2013*).

Thioester-containing proteins (TEPs) have also been suggested to function as pattern recognition proteins, possibly targeting microbes for phagocytosis (*Stroschein-Stevenson et al., 2005*; *Wang and Wang, 2013*). We did not find induction of thioester-containing proteins (TEPs) but rather repression, for instance of *TEP-D* (*Supplementary file 2*). This is surprising, since TEPs are upregulated in *Tribolium* larvae and adults (*Altincicek et al., 2013*; *Behrens et al., 2014*); and *Drosophila* (*Stroschein-Stevenson et al., 2005*; *Wang and Wang, 2013*). Similar to *Drosophila*, however, we did find induction of a putative TEP/complement-binding receptor-like protein (*LpR2*). We also found induction of *C-type lectin 6* and repression of *C-type lectin 1* and *13*. These lectins are thought to be involved in microbial recognition, but no induction or repression has been found in *Drosophila* or *Tribolium* adults (*De Gregorio et al., 2001*; *Altincicek et al., 2013*).

The serine proteases and serpins have significantly expanded in number in *Tribolium* (*Zou et al., 2007*), similar to *Anopheles* (*Christophides et al., 2002*). Interestingly, most of them seem to be functional in the immune response as we found induction of 36 serine proteases and serpins and repression of another 10 upon infection (*Supplementary file 2*). This number is much higher than previously reported for adults (*Altincicek et al., 2013*). Of the Spaetzle ligands, we found induction of *spz1* and *spz2* and repression of *spz4* and *5* (*Supplementary file 2*). In larvae and adults, however, different Spaetzles were induced or repressed, indicating specific use at different stages of the life cycle (*Altincicek et al., 2013*; *Behrens et al., 2014*).

In total, 51 of the 78 immune genes that are differentially regulated in wild-type eggs are involved in bacterial recognition and extracellular signal transduction, showing the prominence of these extracellular processes in the modulation of the immune response of the *Tribolium* egg.

## Transmembrane and intracellular signal transduction

We found induction of several intracellular signaling components of the Toll, IMD, and JNK pathways upon immune challenge of *Tribolium* eggs (*Figure 4A*, *Supplementary file 2*). This suggests that these pathways are largely functionally conserved between *Drosophila* and *Tribolium*, although we could hardly detect expression of *dredd*, the endoprotease that cleaves Relish for nuclear translocation. Similar to larvae and adults (*Altincicek et al., 2013*; *Behrens et al., 2014*), JAK-STAT pathway components were not differentially regulated. Interestingly, we found significant upregulation of the *toll3* receptor upon infection. This was also found in larvae and adults (*Altincicek et al., 2013*; *Behrens et al., 2014*) and suggests that it is not *toll1*, but *toll3* that plays a major role in the innate immune response of *Tribolium*.

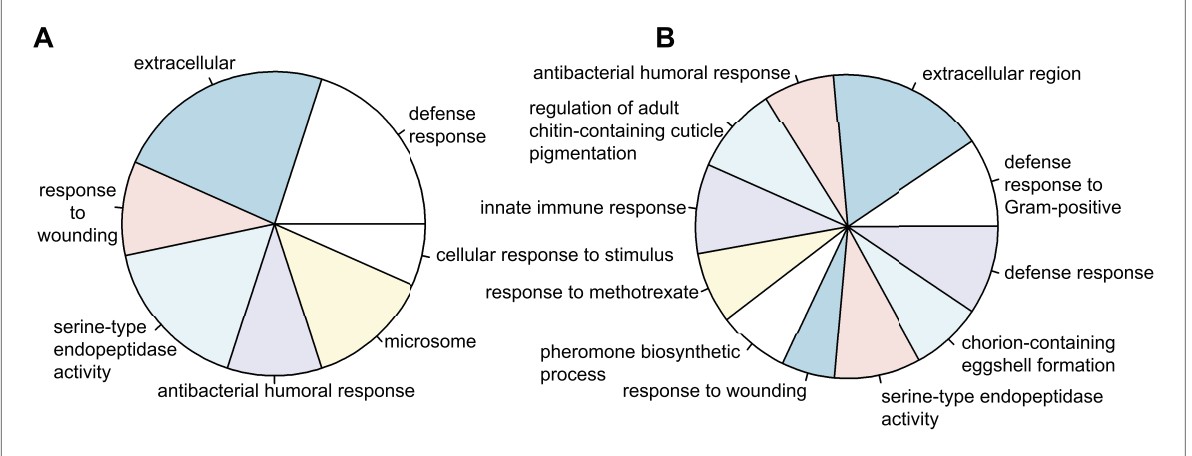

**Figure 3**. Types of genes that are differentially regulated. (**A**) Significantly over-represented GO-terms among the genes induced in wild-type eggs after septic injury (p < 0.001). (**B**) Significantly over-represented GO-terms among the genes induced in control RNAi eggs after septic injury (p < 0.001). These categories indicate that the detected differential regulation does not result from artefacts induced by treatments (such as death or delayed development) and show that *Tribolium* eggs display an elaborate immune response.

## Execution mechanisms

As expected, we found the highest induction amongst the antimicrobial peptides. We detected generally more than 500-fold upregulation of defensins, attacins, coleoptericins, cecropins, and thaumatin (*Figure 4A*, *Supplementary file 2*). This means that *Tribolium* eggs can induce AMPs to comparable levels as larvae and adults (*Altincicek et al., 2013*; *Behrens et al., 2014*). We also found upregulation of *prophenoloxidase1* (*proPO1*), a gene involved in melanisation, and of *heme peroxidase 11*, a *dual oxidase* (*DUOX*) ortholog involved in the production of reactive oxygen species (*Supplementary file 2*). This shows that *Tribolium* eggs are indeed able to respond with the full complement of immune defense mechanisms.

Currently, 19 AMPs are recognized in *Tribolium,* based on homology with known AMPs. However, due to the presence of species-specific AMPs and extreme sequence diversity of these molecules, homology based searches have probably missed several AMPs (*Zou et al., 2007*; *Yang et al., 2011*). AMPs are generally small (less than 30 kDa), cationic, hydrophobic, and possibly have high glycine and/or proline content (*Bulet et al., 2004*; *Bulet and Stöcklin, 2005*). Based on the antimicrobial peptide database (*Wang et al., 2009*), many proteins encoded in the *Tribolium* genome fulfil those criteria and are identified as candidate antimicrobial peptides. Using our RNA sequencing data, however, we could select those candidate proteins that exhibit at least a twofold induction upon infection. Based

**Table 1.** Number of differentially expressed immune genes in *Tribolium castaneum* eggs

| | Wild-type sterile injury | | Wild-type septic injury | | Control sterile injury | | Control septic injury | | *Tc-zen1* sterile injury | | *Tc-zen1* septic injury | |
|---|---|---|---|---|---|---|---|---|---|---|---|---|
| Microbial recognition | 4 | 1 | 7 | 2 | 6 | 0 | 8 | 3 | 1 | 0 | 0 | 0 |
| Extracellular signal transduction and modulation | 27 | 6 | 32 | 10 | 33 | 4 | 34 | 10 | 4 | 5 | 2 | 0 |
| Intracellular transduction pathways (Toll/IMD/JNK/JAK-STAT) | 2 | 1 | 3 | 2 | 2 | 2 | 6 | 3 | 3 | 2 | 2 | 1 |
| Execution/stress | 12 | 0 | 20 | 2 | 16 | 4 | 24 | 7 | 5 | 2 | 3 | 1 |
| **Total** | **45** | **8** | **62** | **16** | **57** | **10** | **72** | **23** | **13** | **9** | **7** | **2** |
| | up | down | up | down | up | down | up | down | up | down | up | down |

Blue = induction, red = repression.

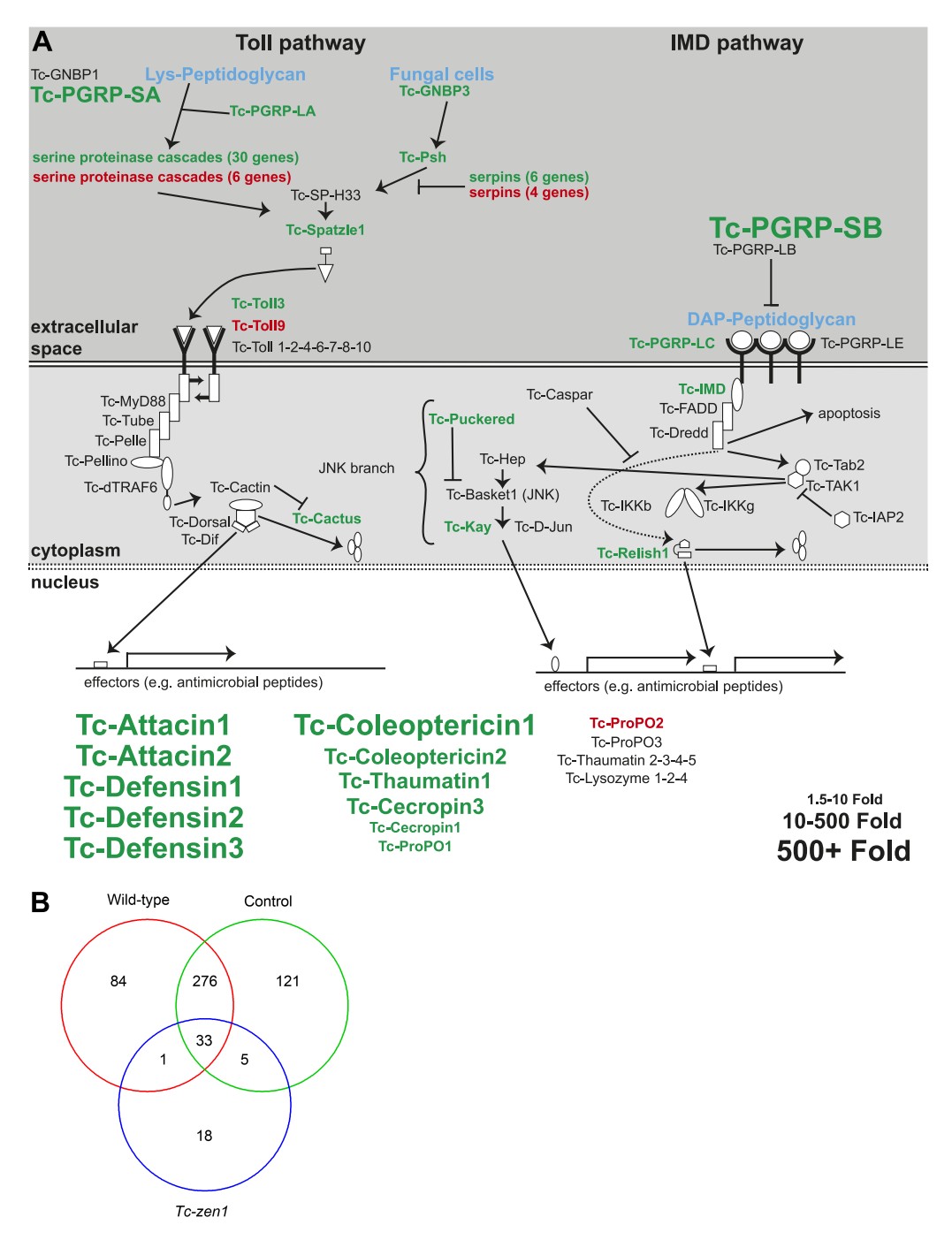

**Figure 4**. Immune-responsive genes in wild-type, control, and *Tc-zen1* RNAi eggs. (**A**) Schematic representation of the immune signaling pathways in *Tribolium* as described in ***Zou et al. (2007)***. Significantly induced genes after septic injury in wild-type or control RNAi eggs are indicated in green; significantly repressed genes after septic injury in wild-type or control RNAi eggs are indicated in red. Genes not differentially expressed are black. The size of the gene names represents the fold change (small = 1.5- to 10-fold, medium = 10- to 500-fold, large = 500 + fold expression). (**B**) Venn diagram showing the number of differentially expressed genes in septically injured eggs as compared to naive eggs (FDR < 0.01). In total, 538 genes are differentially expressed upon infection, of which 394 in wild-type eggs, 435 in control RNAi eggs, and only 57 in *Tc-zen1* RNAi eggs. This means that *Tribolium* eggs display an extensive transcriptional response upon infection and that this response is largely abolished in eggs without a serosa.

on these criteria, we found 20 potential new AMPs (**Table 2**, we included the properties of several known AMPs as a reference). Although the antimicrobial properties of these peptides still have to be experimentally verified, this shows the strength of unbiased approaches to find novel immune genes.

## The immune response is dependent on the extraembryonic serosa

To investigate the role of the serosa in the immune response, we compared the transcriptional response of wild-type and control eggs to the response of serosa-less eggs. Of all 538 genes differentially regulated upon bacterial challenge, 481 genes are only responsive in eggs with a serosa. The vast majority, 276 genes, are differentially regulated in both wild-type and control eggs but not in serosa-less eggs (**Figure 4B**). In the serosa-less *Tc-zen1* RNAi eggs, merely 57 genes are differentially regulated upon microbial challenge, despite our finding that RNAi rather increases the number of immune responsive genes. Of all 368 *Tribolium* genes that are annotated as immune genes (**Zou et al., 2007**; **Altincicek et al., 2013**), only nine were differentially regulated upon infection in serosa-less eggs (**Table 1** and **Supplementary file 2**). Except for *serpin24*, all of the other eight genes were also differentially regulated

**Table 2.** Antimicrobial properties of known and potential new antimicrobial peptides in *Tribolium castaneum*.

| Gene ID | Molecular weight (kDa) | Peptide length (AA) | Hydrophobic ratio | Net charge | Glycine content | Proline content | Fold change wild-type | Fold change control |
|---|---|---|---|---|---|---|---|---|
| Cecropin1/TC000499 | 3.67 | 31 | 58% | +5 | 6% | 0% | Inf | Inf |
| Cecropin3/TC000500 | 9.80 | 90 | 43% | +2 | 6% | 13% | Inf | 49x |
| attacin2/TC007738 | 15.80 | 145 | 37% | +7 | 12% | 4% | 3098x | 2190x |
| Coleoptericin1/TC005093 | 15.99 | 141 | 30% | −1 | 9% | 7% | 2392x | 18067x |
| Defensin2/TC010517 | 8.73 | 79 | 50% | +6 | 6% | 1% | 1183x | Inf |
| Defensin3/TC012469 | 9.42 | 83 | 50% | +7 | 3% | 1% | 908x | Inf |
| attacin1/TC007737 | 17.49 | 165 | 28% | +9 | 18% | 3% | 869x | 3696x |
| TC007858 | 20.14 | 182 | 35% | 0 | 11% | 3% | 484x | 54x |
| Defensin1/TC006250 | 14.91 | 132 | 46% | +11 | 4% | 3% | 187x | 1551x |
| TC011036 | 12.89 | 109 | 39% | +13 | 2% | 6% | 138x | 11x |
| Coleoptericin2/TC005096 | 15.96 | 141 | 30% | −1 | 9% | 7% | 91x | 227x |
| TC015479 | 13.00 | 120 | 42% | +6 | 5% | 1% | 80x | 26x |
| TC007763 | 16.87 | 158 | 37% | +4 | 6% | 17% | 47x | 67x |
| TC004646 | 15.04 | 135 | 34% | +2 | 7% | 7% | 40x | 29x |
| TC008806 | 15.83 | 142 | 33% | +2 | 10% | 2% | 31x | 37x |
| TC009336 | 13.50 | 137 | 30% | −4 | 39% | 2% | 15x | 7x |
| TC014565 | 20.73 | 176 | 38% | +17 | 2% | 2% | 14x | 9x |
| TC001030 | 14.62 | 137 | 29% | +9 | 10% | 12% | 9x | 16x |
| TC001784 | 13.54 | 150 | 27% | +7 | 43% | 2% | 8x | 6x |
| TC005478 | 13.70 | 122 | 45% | +10 | 4% | 1% | 6x | 7x |
| TC015612 | 20.32 | 182 | 36% | +7 | 6% | 6% | 6x | 2x |
| TC007901 | 7.25 | 64 | 25% | +5 | 7% | 10% | 6x | 5x |
| TC015304 | 19.30 | 180 | 38% | +2 | 6% | 9% | 5x | no hit |
| TC011733 | 11.89 | 106 | 46% | +3 | 2% | 0% | 5x | 5x |
| TC003374 | 12.22 | 124 | 61% | +3 | 1% | 9% | 2x | 9x |
| TC008557 | 17.82 | 172 | 31% | +2 | 18% | 0% | 3x | 5x |
| TC015754 | 15.69 | 140 | 34% | +5 | 4% | 7% | 2x | 2x |
| TC000435 | 11.84 | 105 | 37% | +5 | 5% | 0% | 2x | 2x |
| TC009096 | 12.84 | 111 | 16% | +16 | 9% | 6% | 2x | 2x |

In the table are known antimicrobial peptides and those proteins that show at least a twofold induction upon infection, they are smaller than 200 amino acids and are not negatively charged. *TC009336* was included because of the high glycine content.

in response to sterile injury, indicating that they do not respond to infection but to wounding. Notably, none of the AMPs is induced upon infection in serosa-less eggs, neither *proPO1* nor the DUOX ortholog *Hpx11* (*Supplementary file 2*). Thus, the serosa is essential for the early immune response of the *Tribolium* egg.

These data corroborate our previous qPCR study showing that AMP and PGRP upregulation upon infection is abolished in serosa-less eggs (*Jacobs and van der Zee, 2013*). To see if we could also independently confirm serosa-dependent induction of some of our newly identified candidates, we performed qPCR on the transmembrane recognition protein of the IMD pathway *PGRP-LC*, the serine proteases *cS-P8*, *SPH-H57*, *SPH-H70*, the serine protease inhibitors *serpin24* and *serpin26*, the Toll receptor *toll3* and the novel potential AMPs *TC004646*, *TC007763*, *TC007857*, *TC008806*, and *TC015479* (*Figure 5*). The fold-changes detected by qPCR after sterile and septic injury of wild-type eggs match the values found in the RNAseq data. The largest deviation was found for the potential AMP *TC007858* that is upregulated 156 times upon septic injury in our qPCRs but 484 times according to the RNAseq data (*Figure 5J*). Most importantly, all qPCRs convincingly showed the absence of induction in *Tc-zen1* RNAi eggs, thus providing independent support for our conclusion that the serosa is required for the immune response in *Tribolium* eggs.

To investigate if it is the serosal epithelium itself that expresses the identified immune genes and to exclude indirect effects, we performed in situ hybridization on two AMPs (*thaumatin1* and *attacin1*) of which mRNA length permitted in situ detection. In naive eggs, we could not detect *thaumatin1* or *attacin1* expression. In contrast, expression was obvious in challenged eggs (*Figure 6*). In these eggs, brown melanisation was found at the site of injury (asterix in *Figure 6A and A′* and arrowhead in *Figure 6G*) and the individual nuclei of the serosa can be distinguished from the oversaturated DAPI signal marking the germ-band (*Figure 6B,E, H*) (*Handel et al., 2000*). The *thaumatin1* expression clearly associates with the large polyploid serosal nuclei and not with the dense cells of the germ-band (overlay in *Figure 6C and C′*). A deeper focal plane of a different egg demonstrates exclusive expression in the overlying serosa on the outer surface (*Figure 6D,D′*) and not in the underlying embryo proper (*Figure 6E,F*). Also *attacin1* expression consistently associated with the large polyploid serosal nuclei (*Figure 6G–I′*).

Thus, it is the serosal epithelium itself that expresses these AMPs upon infection. Although we cannot exclude an indirect role of the serosa in the expression of the other identified immune genes, we propose that the serosa itself expresses these genes and thus regulates the described immune response involving melanisation, the generation of reactive oxygen species, and the massive production of AMPs.

## The serosa constitutively expresses some immune genes

To discover immune genes that are constitutively expressed in the serosa, we compared the transcriptomes of naive *Tc-zen1* RNAi eggs to naive wild-type eggs. We found 44 immune genes that have serosa-dependent expression (*Table 3*). Of these genes, more than 75% is involved in the recognition of microbes and extracellular signal transduction such as *PGRP-LA*, many serine proteases and *Spz4* and *Spz5* (*Table 3*). In contrast, most of the genes of the intracellular signal transduction were present in *Tc-zen1* RNAi eggs at similar levels as in wild-type eggs. Notably, the transmembrane receptor *toll3* exhibits higher expression in unchallenged eggs with a serosa than in eggs without a serosa. These data indicate that the serosa is an immune competent epithelium that expresses many genes involved in bacterial recognition and transduction of this recognition to receptor activation.

To confirm constitutive expression of these identified genes, we performed in situ hybridization on naive eggs. We chose the receptor *toll3* that shows two times higher expression in eggs with a serosa and the *scavenger receptor B5* that shows 30 times higher expression in eggs with a serosa (*Table 3*). We found ubiquitous expression of *toll3* in the egg (*Figure 7A*). Although *toll3* was clearly expressed in the serosa (partly detached from the egg *Figure 7A′*), we also detected expression in the embryo. As in situ hybridization is not a quantitative technique, and because the serosal cells are flat and thin, it is possible that we could not detect the twofold higher expression in the serosa. For *scavenger receptor B5* that has a 30-fold higher expression in eggs with a serosa, we did find clear expression in the serosal epithelium (*Figure 7D*), whereas the underlying germ-band was not stained (*Figure 7F and F′*). We propose that all genes listed in *Table 3* are constitutively expressed in the serosa and thus make the serosa an immune-competent frontier epithelium.

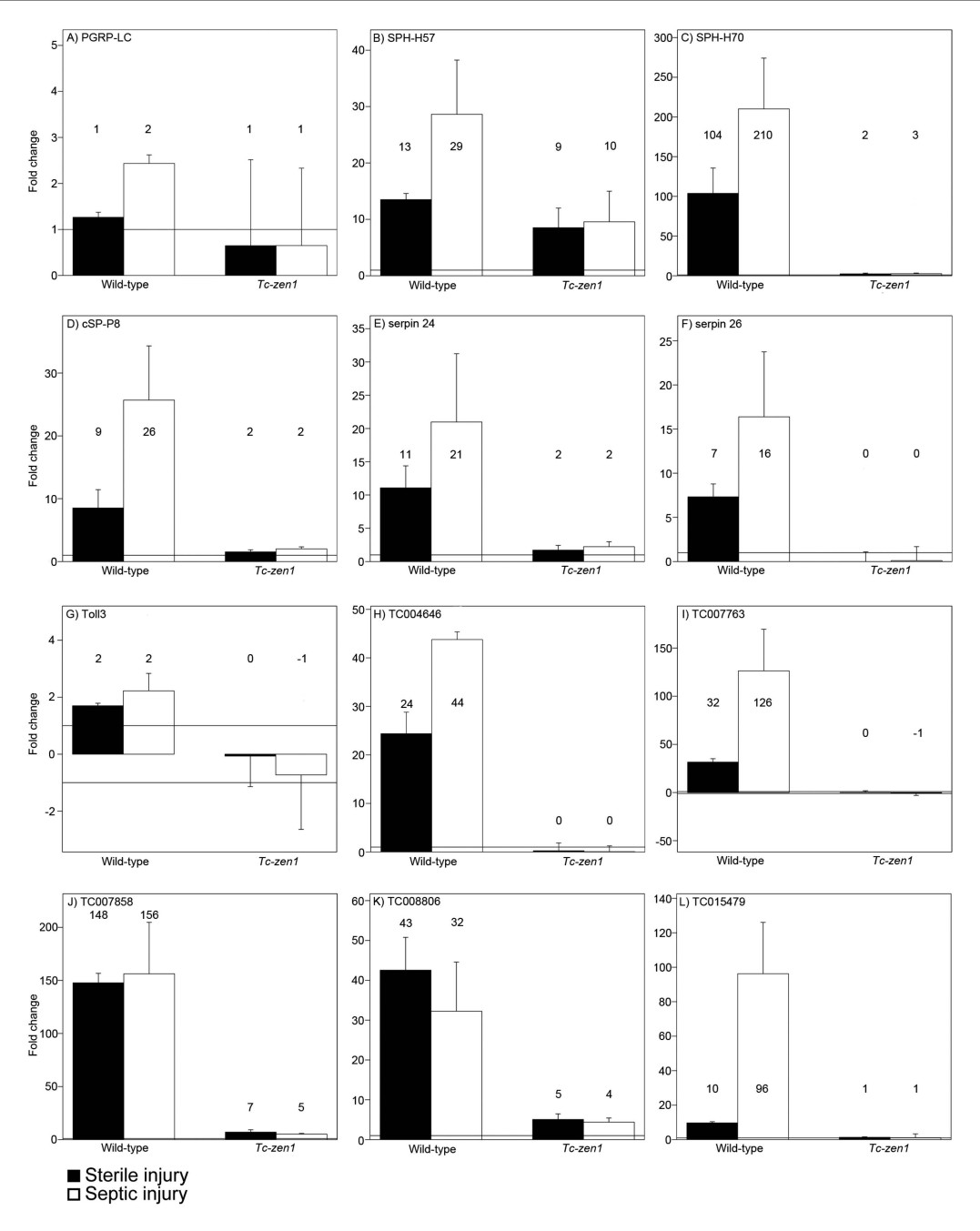

**Figure 5**. RT-qPCR verification of immune gene expression. The expression levels of several immune genes was verified by RT-qPCR. Expression shown relative to the expression in naive eggs, the mean fold change of the biological replicates (based on two technical replicates) is plotted and error bars show the standard error. Black bars represent expression after sterile injury, white bars represent expression after septic injury. Expression levels measured by RT-qPCR show very similar results as the expression levels measured by RNAseq (See **Supplementary file 2**). (**A**) *PGRP-LC*, (**B**) *SPH-H57*, (**C**) *SPH-H70*, (**D**) *cSP-P8*, (**E**) *serpin24*, (**F**) *serpin26*, (**G**) *toll3*, (**H**) *TC004646*, (**I**) *TC007763*, (**J**) *TC007858*, (**K**) *TC008806*, (**L**) *TC015479*. See 'Materials and methods' for experimental details.

Taken together, we have shown that the eggs of the beetle *Tribolium castaneum* display an extensive transcriptional immune response. This response is entirely dependent on the serosa, an extraembryonic epithelium that envelops yolk and embryo. This immune competent frontier epithelium constitutively expresses some immune genes and can induce massive amounts of AMPs.

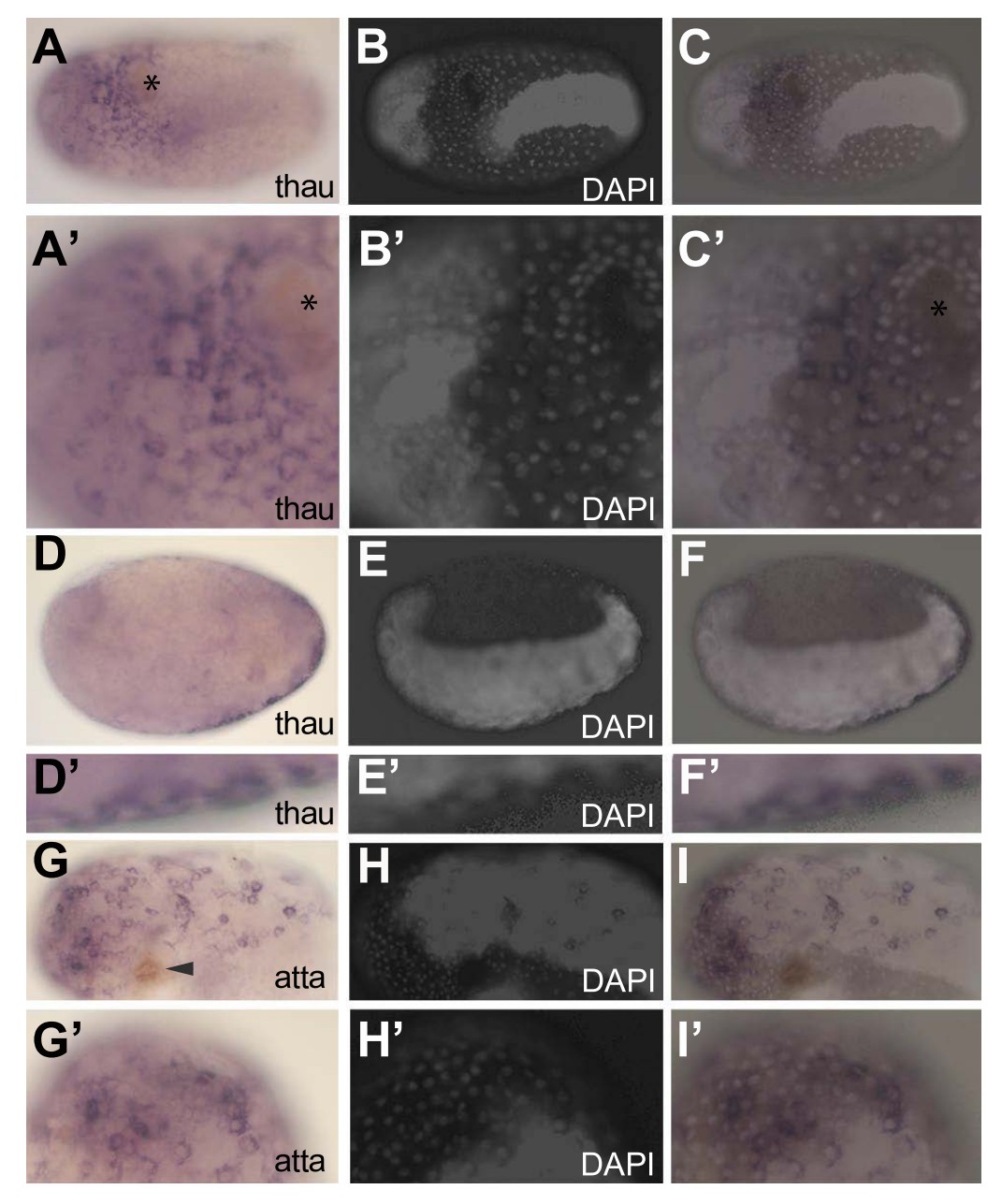

**Figure 6**. In situ hybridization showing expression of AMP genes in the serosa upon septic injury. (**A–F**) *Thaumatin1* in situ hybridization. (**A**) Superficial view. *Thaumatin1* is expressed around the site of injury (asterix). Brown melanisation is observed around the site of injury. (**A′**) Magnification of the expression area shown in (**A**). Asterix marks the site of injury. (**B**) DAPI counterstaining of the same egg as in (**A**). The large polyploid serosal nuclei can be distinguished from the oversaturated DAPI signal from the germ-band. Head lobes to the left. (**B′**) magnification of (**B**). (**C**) Overlay of the in situ hybridization shown in **A** and the DAPI staining shown in (**B**). The *thaumatin1* expression associates with the large polyploid serosal nuclei and is not found in the embryo proper. (**C′**) Magnification of the expression area shown in (**C**). (**D**) Focal plane through the egg. *Thaumatin1* is expressed in a thin outer layer at the surface of the egg. (**D′**) Magnification of the expression area shown in (**D**). (**E**) DAPI staining of the same egg shown in (**D**). The embryo is brightly visible. Head to the left. (**E′**) Magnification of **E**. (**F**) Overlay of the in situ hybridization shown in **D** and the DAPI staining shown in (**E**). (**F′**) Magnification of the expression area. (**G**) *Attacin1* in situ hybridization. Brown melanisation is visible around the site of injury (arrowhead). (**G′**) Magnification of the anterior region of the egg shown in (**G**). (**H**) DAPI staining of the same egg shown in **G**. The germ-band is brightly stained (head to the left) and the separate large serosal nuclei are visible. (**H′**) Magnification of the anterior of the *Figure 6. Continued on next page*

*Figure 6. Continued*

egg shown in (**H**). (**I**) Overlay of the in situ hybridization shown in **G** and the DAPI staining shown in (**H**). *Attacin1* is expressed in the large serosal cells covering the germ-band and is not expressed in the dense cells of the germ band. (**I'**) Magnification of the anterior of the egg shown in (**I**). The *attacin1* staining associates with the large serosal nuclei.

## Discussion

We have provided the first characterization of the complete transcriptional immune response in an insect egg. We identified 538 immune responsive genes in the *Tribolium* egg, of which 481 are only found in eggs with a serosal epithelium. The number of immune-responsive genes found in the *Tribolium* egg is comparable to the number found in larvae (*Behrens et al., 2014*) and higher than what was found in adults (*Altincicek et al., 2013*), but this might be due to differences in sequence coverage. We cannot exclude that some expression differences we found might be due to somewhat delayed development after pricking the eggs. However, the GO-categories of the differentially regulated genes (shown in *Figure 3*) are mainly immune-related, suggesting that an effect of delayed development is negligible.

The induction of several genes from both the Toll and IMD pathway indicates that both pathways are utilized in the immune response of the *Tribolium* egg. It is striking that Toll signaling seems to be involved in innate immunity in the egg, because Toll signaling at this stage has only been associated with developmental functions until now (*Leulier and Lemaitre, 2008*; *Nunes da Fonseca et al., 2008*). In *Drosophila*, Toll1 has been described as the essential immune-related Toll receptor (*Leulier and Lemaitre, 2008*). Other Tolls are not essential for the immune response, except for an antiviral function of Toll7 (*Nakamoto et al., 2012*). Interestingly, *toll3* is significantly upregulated upon infection of the egg and not *toll1*. *Toll3* is also upregulated in infected adults and larvae (*Altincicek et al., 2013*; *Behrens et al., 2014*), suggesting a novel role for *toll3* in *Tribolium* innate immunity. It should be noted that Toll1-4 in *Tribolium* are all closely related to *Drosophila* Toll1 and more distantly to *Drosophila* Toll3 (*Nunes da Fonseca et al., 2008*, *Zou et al., 2007*). Thus, sub-functionalization into developmental and immune-related functions might have occurred among the *Tribolium* Toll1-4 paralogs.

In *Tribolium*, only 19 AMPs have been identified (*Zou et al., 2007*). This is in strong contrast to another beetle species, *Harmonia axyridis*, in which more than 50 putative AMPs have been recognized (*Vilcinskas et al., 2013*). We were able to identify 20 new potential AMPs based on the antimicrobial properties of known AMPs (*Bulet et al., 1999*; *Bulet and Stöcklin, 2005*). Additional AMPs might still be discovered, as we have not investigated peptides longer than 200 amino acids. Some of these long peptides, for instance the Thaumatins, are known to have antimicrobial properties (*Altincicek et al., 2008*, *2013*). We might also have missed AMPs because some might be specifically expressed at other stages, for instance in larvae or adults. Although activity assays against bacteria and fungi are needed to verify antimicrobial properties, the discovery of 20 new potential AMPs shows the power of our experimental strategy for getting an unbiased understanding of insect immunity.

The most important conclusion of our study is that the immune response in *Tribolium* eggs depends on the extraembryonic serosa. To delete the serosa, we used parental *Tc-zen1* RNAi (*van der Zee et al., 2005*). Formally, it is possible that the lack of the immune response we reported is not caused by the absence of the serosa but by a more direct effect of *Tc-zen1* RNAi, for instance if the transcription factor Zen would directly regulate immune genes in the embryo. This is highly unlikely, as *Tc-zen1* is only expressed in the early serosa (*van der Zee et al., 2005*) and is not expressed anymore by the time we performed infection. Indeed, we found only three RNAseq reads that map to *Tc-zen1*, confirming that *Tc-zen1* is practically not expressed at the time we performed experiments. Thus, we are confident to conclude that the lack of the full-range immune response after *Tc-zen1* RNAi is exclusively due to the absence of the serosa.

Eggs with a serosa express crucial bacterial recognition genes, such as *PGRP-LA*, and many extracellular signaling components, such as serine proteases, at higher levels than serosa-less eggs, indicating constitutive expression in the serosa. It could be that these components activate receptors elsewhere in the egg, for instance the *toll3* receptor that is more ubiquitously expressed. However, our in situ hybridizations unambiguously demonstrate that it is the serosal epithelium itself that expresses

**Table 3.** Differentially regulated immune genes in naive wild-type eggs compared to naive *Tc-zen1* RNAi eggs

| Gene ID | Description | Fold change | FDR adjusted p-value | Gene ID | Description | Fold change | FDR adjusted p-value |
|---|---|---|---|---|---|---|---|
| **Extracellular signal transduction and modulation** | | | | | | | |
| TC000247 | cSPH-H2 | 2.70 | <0.01 | TC005754 | serpin22 | 5.26 | <0.01 |
| TC000248 | cSPH-H3 | 4.11 | <0.01 | TC006255 | serpin24 | 0.69 | 0.03 |
| TC000249 | cSPH-H4 | 5.16 | <0.01 | TC011718 | serpin27 | 1.62 | <0.01 |
| TC000740 | SPH-H17 | 9.28 | <0.01 | TC006726 | Spz4 | 3.00 | <0.01 |
| TC000829 | SPH-H18 | 8.26 | <0.01 | TC013304 | Spz5 | 122.56 | <0.01 |
| TC007026 | cSPH-H78 | 29.79 | <0.01 | **Microbial recognition** | | | |
| TC012390 | SPH-H129 | 1.60 | <0.01 | TC002789 | PGRP-LA | 3.95 | 0.02 |
| TC000495 | cSP-P8 | 6.57 | <0.01 | TC014664 | TEP-B | 2.90 | 0.02 |
| TC000497 | cSP-P10 | 4.50 | <0.01 | TC005976 | PSH | 3.43 | <0.01 |
| TC000547 | SP-P13 | 2.41 | <0.01 | TC006978 | C-type lectin1 | 14.52 | <0.01 |
| TC000635 | SP-P16 | 2.54 | <0.01 | TC013911 | C-type lectin 13 | 18.21 | <0.01 |
| TC004160 | cSP-P44 | 9.79 | <0.01 | **Toll-signalling pathway** | | | |
| TC004624 | cSP-P52 | 0.52 | <0.01 | TC004438 | Toll3 | 2.28 | <0.01 |
| TC004635 | cSP-P53 | 51.56 | <0.01 | **IMD-signalling pathway** | | | |
| TC005230 | cSP-P61 | 250.00 | <0.01 | TC014708 | NFAT | 2.01 | <0.01 |
| TC006033 | SP-P68 | 1.54 | <0.01 | **Execution mechanisms** | | | |
| TC009090 | cSP-P91 | 2.80 | <0.01 | TC005375 | hexamerin2 | 0.38 | <0.01 |
| TC009092 | cSP-P93 | 3.00 | <0.01 | TC005493 | Heme peroxidase 1 | 3.84 | <0.01 |
| TC009093 | cSP-P94 | 27.76 | <0.01 | TC015234 | Heme peroxidase 2 | 6.30 | <0.01 |
| TC013277 | cSP-P136 | 3.04 | <0.01 | TC010356 | Scavenger receptor-B13 | 0.60 | 0.03 |
| TC013415 | SP-P141 | 11.85 | <0.01 | TC015854 | Scavenger receptor-B2 | 1.91 | <0.01 |
| TC000760 | serpin1 | 5.29 | <0.01 | TC014946 | Scavenger receptor-B5 | 29.29 | <0.01 |
| TC005750 | serpin18 | 1.92 | <0.01 | TC000948 | Scavenger receptor-B6 | 163.90 | <0.01 |
| TC005752 | serpin20 | 2.30 | <0.01 | TC014954 | Scavenger receptor-B9 | 1.96 | <0.01 |

SP = serine protease; SPH = non-catalytic serine protease; cSP = clip-domain serine protease.

AMPs upon infection, indicating that it is the serosal epithelium itself that harbors the functional immune response reducing bacterial propagation in infected eggs (*Figure 1*).

Overall, bacterial infection of *Tribolium* eggs induces genes involved in melanisation, the acute-phase oxidative response, and AMP production and differentially regulates many other immune genes. This response is completely abolished in eggs without a serosa, the extraembryonic epithelium that envelopes yolk and embryo at the interface with the microbe-rich external milieu. Barrier epithelia like the midgut have recently been highlighted as key players in the local immune defenses in insects (*Davis and Engstrom, 2012*; *Ferrandon, 2013*). We conclude that the serosa is a frontier epithelium that provides the insect egg with a full-range immune response.

Interestingly, the separation of the serosal cells from the germ rudiment is the first morphological distinction that can be made in the blastoderm of the developing egg (*Handel et al., 2000*). The

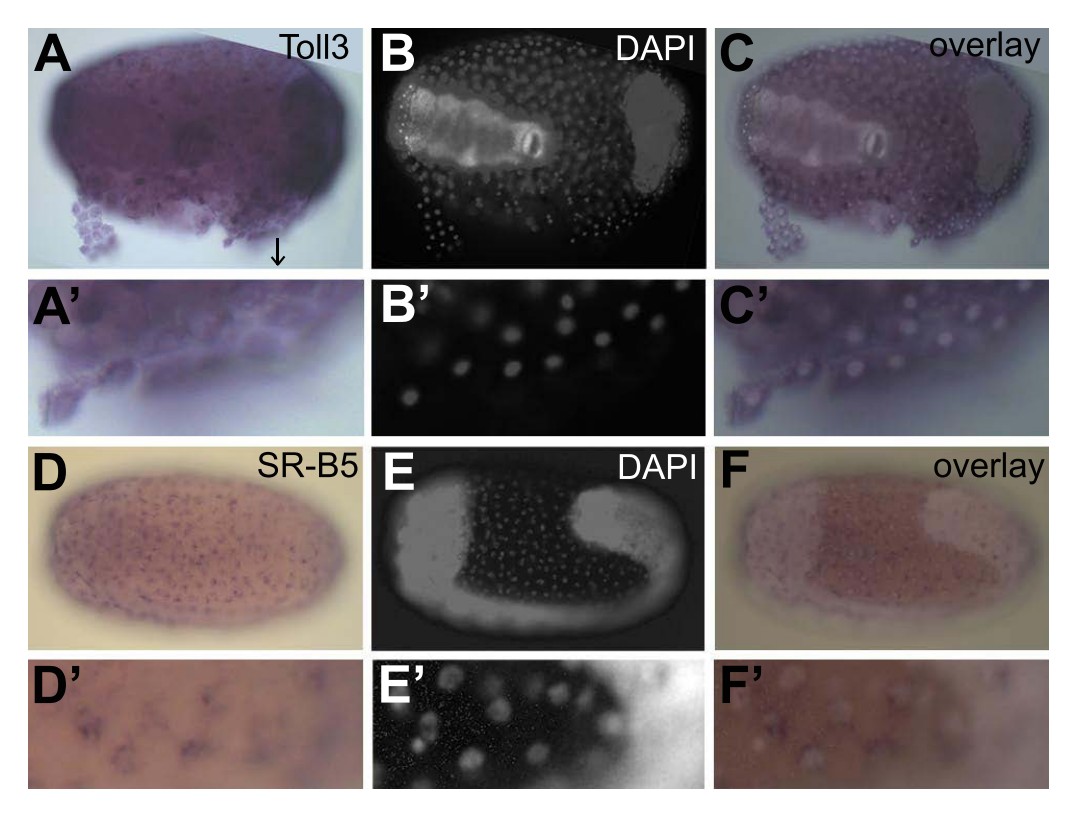

**Figure 7**. Constitutive expression of immune genes in the serosa. (**A–C**) *Toll3* in situ hybridization. (**A**) *Toll3* is expressed in the flat and thin serosal cells (partly detached from the egg) but also in the germ rudiment (head lobes to the right). (**A'**) Magnification of the area indicated with an arrow in (**A**). (**B**) DAPI staining of the same egg shown in (**A**). The bright staining of the germ-band can be distinguished from the large nuclei of the serosa. (**C**) Overlay of the in situ hybridization shown in **A** and the DAPI staining shown in (**B**). (**C'**) Magnification of (**C**). *Toll3* is expressed in cells of the serosa. (**D–F**) *Scavenger receptor B5* in situ hybridization. (**D**) *Scavenger receptor B5* shows expression in every serosal cell at the surface. (**D'**) Magnification of (**D**). (**E**) DAPI staining of the same egg shown in (**D**). The germ-band is brightly stained (head to the left) and the staining of the serosal nuclei is clearly visible when not overwhelmed by staining of the dense nuclei of the germ-band. (**E'**) Magnification of (**E**). The serosal nuclei are visible. Bright staining of the germ-band to the right. (**F**) Overlay of the in situ hybridization shown in **D** and the DAPI staining shown in (**E**). *Scavenger receptor B5* expression follows the serosal nuclei and is not detected in the germ-band. (**F'**) Magnification of (**F**). *Scavenger receptor B5* mRNA is detected around the large polyploid serosal nuclei and not around the dense nuclei of the germ rudiment.

serosal cells will have enveloped the complete embryo before the ectoderm starts to differentiate. These serosal cells can provide the insect egg with an innate immune response long before the embryonic ectoderm or trachea is immune responsive. In addition, the polyploid nuclei allow the serosal cells to quickly synthesize large amounts of proteins providing protection for the vulnerable developing embryo. Thus, the serosa is well suited to provide early immune protection to the egg. *Drosophila* eggs do not develop a serosa, as this extraembryonic membrane was lost in the Schizophoran flies (*Schmidt-Ott, 2000*; *Rafiqi et al., 2008*). A trade-off with developmental speed might have driven the loss of the serosa in these flies living on ephemeral food sources (*Jacobs et al., 2014*). We suggest that the absence of the serosa in the Schizophora accounts for the poor immune response of *Drosophila* eggs. Since all other insects possess a serosa, we propose that early immune competence is a general property of insect eggs.

## Conclusions

*Tribolium castaneum* eggs can mount a full-range innate immune response involving antimicrobial peptides, melanisation, and the production of reactive oxygen species. This response depends entirely on the extraembryonic serosa, an immune competent frontier epithelium that is absent in *Drosophila*.

## Materials and methods

### Beetles and *Tc-zen1* RNAi

The *Tribolium* stock used for this study was the *T. castaneum* wild-type strain, San Bernardino. Stock keeping and *Tc-zen1* RNAi were performed as described in *van der Zee et al. (2005)*. The control dsRNA was synthesized from a 500-bp vector sequence cloned from the pCRII vector (Invitrogen, Waltham, MA, USA) using the primers 5'-TGCCGGATCAAGAGCTACCAA-3' and 5'-TGTGAGCAAAA GGCCAGCAA-3' and has no targets in the *Tribolium castaneum* genome (See also *Jacobs et al., 2013*; *Jacobs and van der Zee, 2013*).

### Infection

Infection experiments were performed as described in *Jacobs and van der Zee (2013)*. 24- to 40-hr old eggs (total developmental time is close to 85 hr) were pricked with a sterile tungsten needle or with a tungsten needle dipped in a concentrated mix of *E. coli* and *M. luteus* cultures (bacteria provided by D Ferrandon, Strasbourg) or were not pricked at all. To allow comparison to the extensive body of work in *Drosophila*, we have used the same strains of *E. coli* and *M. luteus* as are traditionally used in *Drosophila* (*Ferrandon et al., 2007*). 6 hr later, eggs were used for RNA isolation or in situ hybridization.

### Cfu counts

Cfu's were determined directly after infection (t = 0) or 6 hr after infection (t = 6). Eggs were shortly washed for 15 s in a 0.5% hypochlorite solution to sterilize the outside and rinsed with water. 10 eggs were pooled and homogenized in 100 μl water with a sterile pestle. For t = 0, 25 μl of this suspension was directly plated on LB agar plates; for t = 6 these 100 microliters were either diluted 50 times in 50 μl water (for wild-type eggs) or 500 times in 50 μl water (for *Tc-zen1* RNAi eggs). Of these dilutions, 25 μl was plated on LB agar plates. Colonies were counted after an overnight incubation at 37°C, and average numbers of cfu's were calculated per egg. For each combination of time and treatment, the cfu's were measured 10 times. Statistical significance was determined by performing a Pearson's chi-square test. Bacterial load of wild-type eggs increased to on average 32,975 cfu's after 24 hr, but at this time point comparisons to *Tc-zen1* RNAi eggs were unreliable as bacteria might have reached a maximum. At t = 6, bacteria were still in their exponential growth phase and the formula $N_{(t)} = N_{(0)}*e^{kt}$ could be used to calculate the specific growth rate.

### Sample collection for transcriptional analysis

For RNAseq and qPCR, total RNA of approximately 300 eggs was extracted using TRIzol extraction (Invitrogen) after which the RNA was purified and DNA digested on column with the RNeasy kit (Qiagen, Venlo, Netherlands). We collected three biological samples for each of the 9 treatments, giving a total of 27 biological samples (*Figure 2*). cDNA library synthesis and sequencing was performed by the ZF-screens (Leiden, the Netherlands) sequencing company on an Illumina HiSeq2500 sequencer.

### Data analysis and bioinformatics

Sequencing reads were mapped with CLC genomics workbench 6 using the first 51 bp with the highest sequencing quality and score values over 20, allowing 2 mismatches to the reference sequence of the *Tribolium* genome 3.0 which was obtained from Ensemble (*Flicek et al., 2013*). The mismatch cost was set to 2, the insertion cost to 3, the deletion cost to 3, the length fraction to 0.5, and the similarity fraction was set at 0.8. To calculate statistical differences of the expression levels of genes between treatments, we utilized the DESeq package (*Anders and Huber, 2010*) in Bioconductor (*Gentleman et al., 2004*) in R (*R Development Core Team, 2009*). The p values were adjusted for multiple testing with the Benjamini–Hochberg procedure, which determines the false discovery rate (FDR). We trimmed the data to only contain genes that are induced more than twofold or repressed more than twofold. To minimize false discovery rate, we set the cut-off value for significant genes to an FDR of <0.01. DESeq was used to normalize the count data, calculate mean values, fold changes, size factors, variance and p values (raw and adjusted) of a test for differential gene expression based on generalized linear models using negative binomial distribution errors.

### Sequence annotation

Sequence homology searches of predicted reference gene sequences and subsequent functional annotation by gene ontology terms (GO) and InterPro terms (InterProScan, EBI) were determined

using the BLAST2GO software suite v2.6.6 (*Conesa et al., 2005*). First, homology searches were performed through BLASTX against sequences of the *Drosophila* protein database with a cut-off value of 1.0E-10. Subsequently, GO classification annotations were created after which InterPro searches on the InterProEBI web server were performed remotely by utilizing BLAST2GO.

## qPCR

RNA was collected as described under 'Sample collection for transcriptional analysis'. The quality of RNA preparation was confirmed spectro-photometrically and on gel. One microgram of total RNA was used for cDNA synthesis. First strand cDNA was made using the Cloned AMV First Strand Synthesis kit (Invitrogen). Each qRT-PCR mixture (25 µl) contained 2.5 ng of cDNA, and the real-time detection and analyses were done based on SYBR green dye chemistry using the qPCR kit for SYBR Green I (Eurogentec, Seraing, Belgium) and a CFX96 thermocycler (Bio-rad, Hercules, CA, USA). Thermal cycling conditions used were 50°C for 2 min, 95°C for 10 min, then 50 cycles of 95°C for 15 s, 60°C for 30 s, 72°C for 30 s; this was followed by dissociation analysis of a ramp from 65°C to 95°C with a read every 0.5°C. Relative quantification for each mRNA was done using the Livak method (*Livak and Schmittgen, 2001*). The values obtained for each mRNA were normalized by RPL13a mRNA amount for *Tribolium* (Primers as in *Lord et al., 2010*). Total RNA for each treatment was isolated two times (biological replication) and each sample was measured by qRT-PCR twice (technical replication). The primers used for qPCR are in *Supplementary file 3*.

## In situ hybridizations

In situ hybridizations involving alkaline phosphatase-based visualization of DIG-labelled probes were essentially performed as described in *Tautz and Pfeifle (1989)*, but without the proteinase K step. Eggs were fixed for 20 min in a 1:1 mix of heptane and 3.7% formaldehyde in PBST. As the serosa tightly associates with the vitelline membrane, we used *Tc-CHS1* RNAi eggs (*Jacobs et al., 2013*), making it possible to manually dissect eggs containing the serosa from the vitelline membrane. The following primers were used to amplify 500-bp fragments of *thaumatin1*, *attacin1*, *toll3*, and *scavenger receptor B5*.

> *Thaumatin1* FW 5'-CTAAGCGAAGGGGGTTTCGT-3' RV 5'-TTTGTGGTCATCGTAGGCGT-3'
> *Attacin1* FW 5'-ATCGTCCAAGACCAGCAAGG-3' RV 5'-GAAGCGGTGGCTAAACTGGA-3'
> *Toll3* FW 5'-AACTGGGAGGTTTTGCACAC-3' RV 5'-AACTCCATTTTCCCCCAAAC-3'
> *SR-B5* FW 5'-AGCCAGGGAGTTCATGTTCG-3' RV 5'-TGATTTGGTAACGGACGGCA-3'

PCR fragments were cloned into the TOPO II vector (Invitrogen), according to the manufacturer's protocol. From these plasmids, templates for probe synthesis were amplified using M13 primers. DIG-labelled probes were synthesized using the MEGAscript kit (Ambion, Austin, Texas, USA), according to the manufacturer's protocol, but with Roche RNA-labelling mix (Roche, Basel, Switzerland).

## Data access

The data discussed in this publication have been deposited in NCBI's Gene Expression Omnibus (*Barrett et al., 2013*) and are accessible through GEO Series accession number GSE54018 (http://www.ncbi.nlm.nih.gov/geo/query/acc.cgi?acc=GSE54018).

## Acknowledgements

We thank Nora Braak, Romée de Blois, and OL van de Peppel for help with the in situ hybridizations. Heiko Vogel from the Max Planck Institute for Chemical Ecology, Jena, Germany for critically reading the manuscript. MvdZ was funded by NWO VENI grant 863.09.014.

## Additional information

### Funding

| Funder | Grant reference number | Author |
|--------|------------------------|--------|
| Nederlandse Organisatie voor Wetenschappelijk Onderzoek | VENI grant - 863.09.014 | Maurijn van der Zee |

The funder had no role in study design, data collection and interpretation, or the decision to submit the work for publication.

**Author contributions**
CGCJ, MZ, Conception and design, Acquisition of data, Analysis and interpretation of data, Drafting or revising the article; HPS, Conception and design, Analysis and interpretation of data, Drafting or revising the article

## Additional files

### Supplementary files

• Supplementary file 1. Summary statistics for *Tribolium castaneum* transcriptome sequencing analysis.

• Supplementary file 2. Significantly differentially expressed immune genes in wild-type, control, and *Tc-zen1* RNAi eggs.

• Supplementary file 3. Primers used for RT-qPCR.

• Supplementary file 4. DEseq output from wild-type eggs, sterile injury compared to naive eggs.

• Supplementary file 5. DEseq output from wild-type eggs, septic injury compared to naive eggs.

• Supplementary file 6. DEseq output from control RNAi eggs, sterile injury compared to naive eggs.

• Supplementary file 7. DEseq output from control RNAi eggs, septic injury compared to naive eggs.

• Supplementary file 8. DEseq output from *Tc-zen1* RNAi eggs, sterile injury compared to naive eggs.

• Supplementary file 9. DEseq output from *Tc-zen1* RNAi eggs, septic injury compared to naive eggs.

### Major dataset

The following dataset was generated:

| Author(s) | Year | Dataset title | Dataset ID and/or URL | Database, license, and accessibility information |
|---|---|---|---|---|
| Jacobs CGC, Spaink HP, van der Zee M | 2014 | The extraembryonic serosa is a frontier epithelium providing the insect egg with a full-range innate immune response | http://www.ncbi.nlm.nih.gov/geo/query/acc.cgi?acc=GSE54018 | Publicly available at NCBI Gene Expression Omnibus. |

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
