## [Decision Letter]

Thank you for sending your work entitled “The extraembryonic serosa is a frontier epithelium providing the insect egg with a full-range innate immune response” for consideration at *eLife*. Your article has been favorably evaluated by Diethard Tautz (Senior editor) and 2 reviewers, one of whom is a member of our Board of Reviewing Editors.

The Reviewing editor and the other reviewers discussed their comments before we reached this decision, and the Reviewing editor has assembled the following comments to help you prepare a revised submission.

The authors should try to address the comments below as much as they can. Much of it can be addressed by revisions of the text. The broad significance of the results should be better explained to make the study more appealing to a broad audience. The main concern of the reviewers was that the study was largely descriptive. However, the reviewers recognize the fact that initial description is necessarily descriptive but can still enable future mechanistic and biological analyses.

The authors argue that the serosa is an important immune organ in a beetle and back this up with some evidence showing that deletions of the serosa changes bacterial growth and the transcriptional response and they do this using RNAseq. They go further and use in situ hybridization to show that some potentially antimicrobial transcripts are made in this tissue.

The result that the serosa is important for the transcriptional response to an infection in this beetle has been published already by these same authors (Jacobs and Van der Zee 2013), although the past work used QRTPCR and not RNAseq. This manuscript is stronger than the past one because this includes some work on microbe growth but that doesn't seem like enough. What is the message in this manuscript that makes it important to be published in a broad biology journal that is aiming for a high impact? Tell me why I should care about the immune response in beetle eggs. I want the authors to tell me the biological stories that make this important.

I thought that the analysis of bacterial growth was poor and quantitation was poor in general. It is hard to write a story about gene induction if you don't have anything more to go on for function than annotation. The result here is better than many but it doesn't have a lot of information that couldn't be gathered simply from a table.

Specific comments are below:

“Epithelia and hemocytes play the main role in local immune defenses that comprise melanization, local AMP production and encapsulation”. This is true but it is like saying that T cells play the main role in T cell responses. The authors have highlighted the processes that are driven by these tissues. My point is that it doesn't seem necessary to push epithelia this hard.

“When microbes enter the fly, they are bound by recognition proteins...” Is this how it works? Is there evidence that it is the actual microbes that are bound or is it the free peptidoglycans released from dividing bacteria that induce the immune response? I thought it was the latter.

There is more to an immune response than up-regulation; there is considerable down regulation upon immune activation and it would be good to mention these things as well.

When discussing what various insects and model organisms have taught us it might be useful to point out the purpose of the studies. Drosophila has been a useful immunological tool because it provoked those working on vertebrate immunity to work on Toll. It has also been useful in scaffolding other insect immune systems and I think that people understand that these various immune systems differ some from each other. What is the purpose of studying the beetle? What has the beetle taught us that is of interest to those that work on beetles, other insects and immunology in general? To get published in a broad biological journal, it is important to have a broad message.

Thanks for mentioning that there are not black and white rules for Gram positive versus Gram negative signaling in Drosophila. It might be useful to just discuss this in terms of elicitors, which is what the host reacts to instead of mentioning the Gram status at all. Why should we expect that a 19th century description of microbiology would be useful in distinguishing signaling mechanisms in the 21st century?

Regarding the activity of the immune response in eggs, there is some recent work from Will Wood's lab that could be cited about the maturation of the immune response in relation to ecdysone signaling.

What is the evidence that “eggs are constantly threatened by pathogens”? When I worked as a developmental biologist I was frustrated with the difficulty of getting anything into a Drosophila egg. Entry isn't easy. Maybe the fly has adopted an alternative response to infection by making impermeable eggs. Why invest in an inducible immune response if you don't need it? I ask because we are often get the argument that insects need a strong immune response because they live in such a dirty environment and the argument makes little sense in light of our developing understanding of our own microbiota. We are filled with microbes and mostly this is good for us. I fear that immunologists often confuse microbes with pathogens.

The lack of attention for immunity in the egg might be explained by the response in Drosophila eggs but it could also be due to a lack of a strong biological question. If it was clear that there were diseases that people cared about that were transmitted through the egg this would drive more research.

“Eggs of this main model organism hardly upregulate AMPS...” Please be quantitative. There is a serious problem in this field where workers substitute relativistic adjectives for quantitation. “Hardly upregulate” is a relative term that reflects more what the authors’ thoughts on the result than what the fly thought of the result. Did the upregulated AMPS do what they were supposed to do? Did they clear bacteria? If they did then the fly is being efficient. Alas, microbe loads and changes in microbe loads were not measured in that paper. There is another problem: often people report immune inductions without reporting the amount of elicitor injected into the host. Immune induction only makes sense when considered as a response to microbes. In the cited paper, the microbe loads weren't measured for the injections and no comparison was made between the amount of bacteria injected into the eggs of the two insects. Eggs from different species have different properties; how can you be sure that the eggs are getting the same amount of material when you are just stabbing them with a dirty needle? I wouldn't be willing to conclude that the cited paper shows a quantitative difference in gene expression.

“…and this absence might account for the poor immune response in Drosophila eggs” I won't win this, but I would love it if immunity wasn't just used as a word to describe the simple transcriptional response of a host but rather it was used to functionally describe the response of the host and then, of course, to look at the effects on the fitness of the host. Do fly eggs kill injected bacteria? That is the real measure of the immune response and not whether some arbitrary marker that hasn't been tested for its antimicrobial activity on the injected microbe is modulated.

Did the flies in this experiment carry Wolbachia? I ask because the authors state that the eggs did not contain many bacteria but they assayed this by culture methods designed to look at enteric human pathogens and not Drosophila native microbiota. Fly eggs are often decorated with a little drop of feces that is full of Lactobacilli and Acetobacter. These won't show up easily on LB plates grown at 37C. I don't know what the native microbiota is for these beetles or whether the beetles use this egg-pooping trick but it makes me wonder.

I am pleased that the authors are finally including some microbe counts in their experiments. I wish that they would use microbes other than *E. coli* and *M. luteus*. Why were these chosen? What biological relevance do they have? One of the reasons to switch to other insects is to use their natural history to ask interesting experiments. It seems sad to repeat the mistakes that were made when testing the immune response of Drosophila.

“These numbers increased to 747 +/- cfu's.” I don't understand this - +/- what? This is usually followed by a variance. I would be much more comfortable with the growth dynamics if the authors would use more than one timepoint. Presumably these bacteria are growing exponentially in the beginning and the authors have looked at less than one doubling. The formula used for bacteria growth is for exponential growth but the authors have only shown a single timepoint and thus can't tell us anything about the shape of the curve. Wouldn't a logistic function be a better idea? The bacteria presumably aren't going to grow forever and will hit a ceiling. True, the beginning of the growth curve will be exponential but if the authors haven't provided the growth curve, how do we know that they are at the beginning? It could be that these two types of flies simply have different maximum loads of bacteria and the bacteria have already reached their maxima. The authors are making a lot of unsubstantiated assumptions about bacterial growth here and it would be easy to solidify this work.

The authors treat the parents with RNAi and knock down a tissue. Is there a control for the effects of an RNAi response on the eggs where tc-zen1 is not knocked down? I ask because I'm worried about the following artifact – RNAi induces an immune response in insects. The authors show that the RNAi induction has an impact on the transcriptional profile of the eggs yet they didn't follow this up with controls. Immune responses have costs. The cost could be that the mother does not endow the egg with the same supplies because the mother is busy fighting a perceived infection. Thus eggs from RNAi treated mothers might have poor immune responses that are poor regardless of what gene was knocked down. I'm making this up but it is plausible – things like this do happen and it is a simple control. Without this control the authors conclude that their knockdown is responsible for the phenotype when it could be that any knockdown produces this phenotype.

One can determine based on the sequence whether a PGRP is likely to be enzymatically active. Which of the Tribolium PGRPs fall into this group? If the authors want to suggest that the expressed PGRPs are effectors, they should do this preliminary legwork.

“We also found induction of PGRP-LB but this was not significant” Then they didn't find induction of PGRP-LB. They should not report insignificant results.

How do the authors propose that the immune response is turned on? Is it a switch and a mere whiff of bacteria turns on the response or is it dose responsive? If dose responsive, is it linear or does it have some other shape. I ask because the authors did not report microbe loads for their immune inductions and we expect them to be different. The results they see could be even larger than they reported as there are 10x more bacteria in the knockdown eggs and low gene expression, therefore the ratio of gene expression to dose is 10x larger than the effects they see without including microbe load.

“We chose the receptor Toll3” Is this an immune signaling Toll? Drosophila has many Tolls and it isn't clear whether more than 2 or 3 have immune functions. Some that had been written off as being non-immune have immune functions but we really don't know. It is risky to make assumptions about a gene without a functional test.

---

## [Author Response]

*The authors should try to address the comments below as much as they can. Much of it can be addressed by revisions of the text. The broad significance of the results should be better explained to make the study more appealing to a broad audience. The main concern of the reviewers was that the study was largely descriptive. However, the reviewers recognize the fact that initial description is necessarily descriptive but can still enable future mechanistic and biological analyses*.

We agree that this paper is not mechanistic. We disagree that the paper is merely descriptive, though. To analyze the function of the serosa, we removed it using *Tc-zen1* RNAi. Our paper does provide functional analysis of the serosa and includes counts of bacterial loads in serosa-less eggs. Our paper is certainly experimental.

*The authors argue that the serosa is an important immune organ in a beetle and back this up with some evidence showing that deletions of the serosa changes bacterial growth and the transcriptional response and they do this using RNAseq. They go further and use in situ hybridization to show that some potentially antimicrobial transcripts are made in this tissue*.

*The result that the serosa is important for the transcriptional response to an infection in this beetle has been published already by these same authors (Jacobs and Van der Zee 2013), although the past work used QRTPCR and not RNAseq. This manuscript is stronger than the past one because this includes some work on microbe growth but that doesn't seem like enough. What is the message in this manuscript that makes it important to be published in a broad biology journal that is aiming for a high impact? Tell me why I should care about the immune response in beetle eggs. I want the authors to tell me the biological stories that make this important*.

In the revised version, we now better explain the ecological relevance of an immune response in insect eggs. The reviews we already cite treat a very extensive body of evidence for infections in insect eggs and their antibacterial activity. In our revised manuscript, we now cite 6 extra primary research papers with concrete examples. Furthermore, we make clearer that, until now, immuno-ecologists have been thinking in terms of maternal investments in antimicrobial activity of eggs. Our paper could shift this paradigm towards attention for the zygotic investment. Furthermore we stress the importance of choosing a model organism that possesses a serosa if one wants to study the immune response in an egg. Drosophila is not representative for insects in this respect.

*I thought that the analysis of bacterial growth was poor and quantitation was poor in general. It is hard to write a story about gene induction if you don't have anything more to go on for function than annotation. The result here is better than many but it doesn't have a lot of information that couldn't be gathered simply from a table*.

We agree that we could present the data in a table. We think, however, that a more visual representation makes the data easier to grasp, especially for non-specialist readers reading *eLife.*

*Specific comments are below*:

*“Epithelia and hemocytes play the main role in local immune defenses that comprise melanization, local AMP production and encapsulation”. This is true but it is like saying that T cells play the main role in T cell responses. The authors have highlighted the processes that are driven by these tissues. My point is that it doesn't seem necessary to push epithelia this hard*.

This is the second sentence of our manuscript. Although this might be a trivial sentence to the specialist reader, we do have the feeling that we should set the frame for the broad readership of *eLife*. We do not have the feeling that we push epithelia any stronger than hemocytes by writing this. We have deleted the word “epithelium” from the concluding sentence of the summary.

*“When microbes enter the fly, they are bound by recognition proteins...” Is this how it works? Is there evidence that it is the actual microbes that are bound or is it the free peptidoglycans released from dividing bacteria that induce the immune response? I thought it was the latter*.

We corrected this inaccuracy in our revised version.

*There is more to an immune response than up-regulation; there is considerable down regulation upon immune activation and it would be good to mention these things as well*.

Throughout the manuscript, we speak about differentially regulated genes. In the originally submitted manuscript, we explicitly mention “induced or repressed genes”. Regarding treating expression of TEP-D, we mention that it is repressed after infection. We also describe repression of 10 serine proteases or serpins. Then we treat downregulation of Spz4 and Spz5 upon infection. As described in its figure legend, we indicate repressed genes in red in Figure 4. Table 1 has separate columns for down regulated genes. Thus, we obviously do treat downregulated genes.

*When discussing what various insects and model organisms have taught us it might be useful to point out the purpose of the studies. Drosophila has been a useful immunological tool because it provoked those working on vertebrate immunity to work on Toll. It has also been useful in scaffolding other insect immune systems and I think that people understand that these various immune systems differ some from each other. What is the purpose of studying the beetle? What has the beetle taught us that is of interest to those that work on beetles, other insects and immunology in general? To get published in a broad biological journal, it is important to have a broad message*.

In our revised version, we now explain what we have learnt from studying other systems. Many biochemical details of pathway activation were for instance uncovered in the beetle *Tenebrio* from which large quantities of hemolymph can be collected. We also explain better that for studying the immune response in eggs, it is important to choose a model insect that possesses a serosa. This is more representative for insects.

*Thanks for mentioning that there are not black and white rules for Gram positive versus Gram negative signaling in Drosophila. It might be useful to just discuss this in terms of elicitors, which is what the host reacts to instead of mentioning the Gram status at all. Why should we expect that a 19th century description of microbiology would be useful in distinguishing signaling mechanisms in the 21st century*?

In our revised version, we now explicitly mention the elicitors in the first paragraph of the introduction.

*Regarding the activity of the immune response in eggs, there is some recent work from Will Wood's lab that could be cited about the maturation of the immune response in relation to ecdysone signaling*.

We cite this paper now.

*What is the evidence that “eggs are constantly threatened by pathogens”? When I worked as a developmental biologist I was frustrated with the difficulty of getting anything into a Drosophila egg. Entry isn't easy. Maybe the fly has adopted an alternative response to infection by making impermeable eggs. Why invest in an inducible immune response if you don't need it? I ask because we are often get the argument that insects need a strong immune response because they live in such a dirty environment and the argument makes little sense in light of our developing understanding of our own microbiota. We are filled with microbes and mostly this is good for us. I fear that immunologists often confuse microbes with pathogens*.

Besides the two excellent and extensive reviews we cite, we now cite extra papers that show the deleterious effect of infections in eggs, for instance by pathogenic fungi or *Serratia* bacteria that are able to penetrate eggs.

*The lack of attention for immunity in the egg might be explained by the response in Drosophila eggs but it could also be due to a lack of a strong biological question. If it was clear that there were diseases that people cared about that were transmitted through the egg this would drive more research*.

We removed this sentence in our revised manuscript.

*“Eggs of this main model organism hardly upregulate AMPS...” Please be quantitative. There is a serious problem in this field where workers substitute relativistic adjectives for quantitation. “Hardly upregulate” is a relative term that reflects more what the authors’ thoughts on the result than what the fly thought of the result. Did the upregulated AMPS do what they were supposed to do? Did they clear bacteria? If they did then the fly is being efficient. Alas, microbe loads and changes in microbe loads were not measured in that paper*.

We now mention the exact fold upregulation in fly eggs and adults. Furthermore, we cite an excellent paper from the Wood laboratory nicely showing that stage 11 *Drosophila* embryos cannot contain an infection of non-pathogenic bacteria leading to reduced survival.

*There is another problem: often people report immune inductions without reporting the amount of elicitor injected into the host. Immune induction only makes sense when considered as a response to microbes. In the cited paper, the microbe loads weren't measured for the injections and no comparison was made between the amount of bacteria injected into the eggs of the two insects. Eggs from different species have different properties; how can you be sure that the eggs are getting the same amount of material when you are just stabbing them with a dirty needle? I wouldn't be willing to conclude that the cited paper shows a quantitative difference in gene expression*.

Indeed, we can’t be sure that we introduced the same number of bacteria in *Drosophila* and *Tribolium* in our previous paper. In this manuscript, we show that we introduce around 50 bacteria into the eggs of *Tribolium*. To the same “stabbing with a dirty needle”-approach, *Drosophila* eggs do not respond. OK, as this reviewer criticizes: we could have introduced fewer bacteria. However, the Woods laboratory has published an experiment in stage 11 *Drosophila* eggs introducing a number of bacteria in the order of magnitude of 10^3^. Still, there was no transcriptional response of AMPs. Thus, I think we can confidently conclude that the response in *Drosophila* is poor.

*“…and this absence might account for the poor immune response in Drosophila eggs” I won't win this, but I would love it if immunity wasn't just used as a word to describe the simple transcriptional response of a host but rather it was used to functionally describe the response of the host and then, of course, to look at the effects on the fitness of the host. Do fly eggs kill injected bacteria? That is the real measure of the immune response and not whether some arbitrary marker that hasn't been tested for its antimicrobial activity on the injected microbe is modulated*.

No, the Will Wood laboratory has shown that stage 11 *Drosophila* eggs cannot contain an infection and die of even non-pathogenic bacteria. We cite this paper in our revised version.

*Did the flies in this experiment carry Wolbachia? I ask because the authors state that the eggs did not contain many bacteria but they assayed this by culture methods designed to look at enteric human pathogens and not Drosophila native microbiota. Fly eggs are often decorated with a little drop of feces that is full of Lactobacilli and Acetobacter. These won't show up easily on LB plates grown at 37C. I don't know what the native microbiota is for these beetles or whether the beetles use this egg-pooping trick but it makes me wonder*.

No, remarkably, *Tribolium castaneum* does not carry *Wolbachia* (Chang and Wade, 1996). But we agree with the reviewer. In the revised version we now explicitly write: “...did hardly contain bacteria that grow on LB agar plates”.

*I am pleased that the authors are finally including some microbe counts in their experiments. I wish that they would use microbes other than E.coli and M. luteus. Why were these chosen? What biological relevance do they have? One of the reasons to switch to other insects is to use their natural history to ask interesting experiments. It seems sad to repeat the mistakes that were made when testing the immune response of Drosophila*.

These bacteria were obtained from the laboratory of Dominique Ferrandon in Strasbourg. These bacteria have traditionally been used for studying the immune response in *Drosophila*. An extensive body of work has been done using these bacteria. We understand that this might be a “mistake” in some sense, but we set out to compare the immune response in *Tribolium* to *Drosophila*. We think that using the exact same strains is required to compare the responses. We added this to the Materials and methods in the revised version.

*“These numbers increased to 747 +/- cfu's.” I don't understand this - +/- what? This is usually followed by a variance. I would be much more comfortable with the growth dynamics if the authors would use more than one timepoint. Presumably these bacteria are growing exponentially in the beginning and the authors have looked at less than one doubling. The formula used for bacteria growth is for exponential growth but the authors have only shown a single timepoint and thus can't tell us anything about the shape of the curve. Wouldn't a logistic function be a better idea? The bacteria presumably aren't going to grow forever and will hit a ceiling. True, the beginning of the growth curve will be exponential but if the authors haven't provided the growth curve, how do we know that they are at the beginning? It could be that these two types of flies simply have different maximum loads of bacteria and the bacteria have already reached their maxima. The authors are making a lot of unsubstantiated assumptions about bacterial growth here and it would be easy to solidify this work*.

Although the error bars are indicated in the graph, we erroneously omitted the exact numbers from the text. We added these numbers to the figure legend in the revised version. Indeed, from the presented data no conclusions can be drawn on the shape of the curve. However, we have data on the bacterial load after 24 hours. This is on average 32,975. We left these data out of the manuscript for the exact concern this reviewer raises: at that point the bacteria might have hit a ceiling. But it makes us confident that after 6h, bacteria are still in their exponential phase. We have added this line of reasoning to the materials and methods. A little bit of calculation supports this assumption: in *Tc-zen1* RNAi eggs, the bacterial load increases from 49 to 7260 in 6 hours. This requires more than 7 divisions. That means slightly more than one division per hour. This is actually close to what bacteria can reach in LB medium, and likely is exponential.

*The authors treat the parents with RNAi and knock down a tissue. Is there a control for the effects of an RNAi response on the eggs where tc-zen1 is not knocked down? I ask because I'm worried about the following artifact – RNAi induces an immune response in insects. The authors show that the RNAi induction has an impact on the transcriptional profile of the eggs yet they didn't follow this up with controls. Immune responses have costs. The cost could be that the mother does not endow the egg with the same supplies because the mother is busy fighting a perceived infection. Thus eggs from RNAi treated mothers might have poor immune responses that are poor regardless of what gene was knocked down. I'm making this up but it is plausible – things like this do happen and it is a simple control. Without this control the authors conclude that their knockdown is responsible for the phenotype when it could be that any knockdown produces this phenotype*.

Yes, we included a control RNAi. It is also clearly indicated in Figure 2 outlining the experimental set-up. We even discuss this issue in the originally submitted manuscript, explaining that the effect of RNAi alone leads to a larger number of differentially regulated genes after infection. Yet, knockdown of *Tc-zen1* using RNAi leads to a dramatic reduction of differentially regulated genes. This makes us confident to conclude that it is this specific knockdown of this gene that causes the phenotype and not any knockdown.

In the main text of the revised version, we now explicitly mention that this control RNAi uses a 500 bp vector sequence without target in the *Tribolium castaneum* genome. We added the details to the Materials and methods.

*One can determine based on the sequence whether a PGRP is likely to be enzymatically active. Which of the Tribolium PGRPs fall into this group? If the authors want to suggest that the expressed PGRPs are effectors, they should do this preliminary legwork*.

PGRP-SB possesses all the characteristic residues for catalytic PGRPs: His 42, Tyr78, His152, Thr158 and Cys160. We added this to the main text. PGRP-SA does not have the Zinc-binding His152 and Cys160.

*“We also found induction of PGRP-LB but this was not significant” Then they didn't find induction of PGRP-LB. They should not report insignificant results*.

In our revised version, we have deleted this sentence.

*How do the authors propose that the immune response is turned on? Is it a switch and a mere whiff of bacteria turns on the response or is it dose responsive? If dose responsive, is it linear or does it have some other shape. I ask because the authors did not report microbe loads for their immune inductions and we expect them to be different. The results they see could be even larger than they reported as there are 10x more bacteria in the knockdown eggs and low gene expression, therefore the ratio of gene expression to dose is 10x larger than the effects they see without including microbe load*.

I don’t see completely what this reviewer is aiming at. We are pleased that this reviewer thinks that the effect of the serosa could be even larger, but in our eyes complex corrections for bacterial loads are speculative and we prefer sticking to our conservative and clear conclusions.

*“We chose the receptor Toll3” Is this an immune signaling Toll? Drosophila has many Tolls and it isn't clear whether more than 2 or 3 have immune functions. Some that had been written off as being non-immune have immune functions but we really don't know. It is risky to make assumptions about a gene without a functional test*.

We chose Toll3 because it was the only Toll receptor that showed a higher expression in eggs with a serosa. This is mentioned 4 lines before in the manuscript. We do not know its function. Our RNAseq paper identifies lots of genes for further research; it is beyond the scope of this paper to investigate them all functionally. We only speculate that Toll3 might have an immune function, as it is also upregulated in response to infection. This was independently found in larvae and adults by other groups.